# Anxiety, Motives, and Intention for Physical Activity during the Italian COVID-19 Lockdown: An Observational Longitudinal Study

**DOI:** 10.3390/ijerph19084689

**Published:** 2022-04-13

**Authors:** Federica Galli, Francesco Giancamilli, Tommaso Palombi, Jacopo Antonino Vitale, Stefano Borghi, Alessandra De Maria, Elisa Cavicchiolo, Pierluigi Diotaiuti, Antonio La Torre, Arnaldo Zelli, Fabio Lucidi, Roberto Codella, Andrea Chirico

**Affiliations:** 1Department of Movement, Human and Health Sciences, University of Rome, “Foro Italico”, 00135 Rome, Italy; federica.galli@uniroma1.it (F.G.); de.maria.alessandra@gmail.com (A.D.M.); arnaldo.zelli@uniroma4.it (A.Z.); 2Department of Psychology of Development and Socialization Processes, “Sapienza”, University of Rome, 00185 Rome, Italy; francesco.giancamilli@uniroma1.it (F.G.); tommaso.palombi@uniroma1.it (T.P.); fabio.lucidi@uniroma1.it (F.L.); 3IRCSS Istituto Ortopedico Galeazzi, 20161 Milan, Italy; jacopo.vitale@grupposandonato.it (J.A.V.); antonio.latorre@unimi.it (A.L.T.); 4Department of Biomedical Sciences for Health, Università degli Studi di Milano, 20133 Milan, Italy; stefano.brg94@gmail.com (S.B.); roberto.codella@unimi.it (R.C.); 5Department of Human, Philosophical and Educational Sciences, University of Salerno, 84084 Fisciano, Italy; ecavicchiolo@unisa.it; 6Department of Human Sciences, Society and Health, University of Cassino and Southern Lazio, 03043 Cassino, Italy; p.diotaiuti@unicas.it; 7Department of Endocrinology, Nutrition and Metabolic Diseases, IRCCS MultiMedica, 20138 Milan, Italy

**Keywords:** COVID-19 lockdown, physical activity, motivation, anxiety, moderated mediation model, Italy

## Abstract

Background: The coronavirus (COVID-19) pandemic led governments to adopt strict containment measures to avoid spreading the virus. These essential measures led to home confinement that influenced both the physical and mental health of populations. Physical activity plays a key role in preventing chronic diseases and promoting protective psychological factors. In the context of a lockdown, understanding the motives that guide people to enact physical activity is an important issue for public health. The present study aimed to evaluate the relation between autonomous motivation and physical activity, considering the role of behavioral intention and anxiety in a longitudinal moderated mediation model. Methods: Italian participants (N = 86; mean_age_ = 29.74, standard deviation = 9.74; female = 53.5%) completed a booklet composed of different questionnaires (motivation, intention, anxiety, and physical activity) 3 weeks apart. Results: The hypothesized model is supported by the evidence; both autonomous motivation and intention are direct predictors of physical activity. The results also show that the direct effect of autonomous motivation on physical activity is stronger in participants with low anxiety, while high levels of anxiety are a significant moderator of the intention–behavior relation. Conclusions: In conclusion, a multidisciplinary approach should be promoting methods and infrastructures to permit people to adhere to physical activity, as a front line against any health emergency.

## 1. Introduction

The new coronavirus disease (COVID-19) was identified in Wuhan, China in December 2019, and in March 2020, the World Health Organization (WHO), officially declared the new COVID-19 a pandemic [1]. According to the reports by the Italian Ministry of Health [2], at the beginning of the pandemic (12 March 2020) Italy had 15.113 people infected and 1.016 deaths. By the end of March, the number of people infected increased dramatically to 105.792, and the number of deaths was 12.428. Consequently, the Italian government adopted strict containment measures for nearly two months to avoid spreading the virus, and a possible collapse of the Italian health care system (i.e., the so-called “phase 1” of the COVID-19 outbreak) [3]. During this “first” lockdown, many activities, considered non-essential, were curbed, and social confinement was mandatory. The restrictive measures, albeit essential to curb the number of virus infections, led, inevitably, to general health issues and home confinement. Altogether, this had a large impact on several aspects of human physical and mental health [4,5]. A prolonged period of staying home is associated with sedentary lifestyles (e.g., many hours spent in smart working, watching television, or using electronic devices), sleep disruption and an altered sleep-wake cycle, modified diet pattern and timing, higher levels of depression, stress, and anxiety, and with some sports restrictions too, e.g., [4,6]. No group activities, in any form, were allowed and non-compliance with this regulation was considered a severe law violation [7]. Unfortunately, even a few days without exercise can reverse the positive physiological adaptations gained through chronic physical training, and the body tends to return to a baseline level very quickly i.e., detraining effect [8]. In other words, physical inactivity has such a negative impact on overall health that just few days of sedentary behaviors are accompanied by several negative consequences, such as muscle loss [9] and a decreased aerobic capacity [10]. Indeed, exercise plays a key role in the prevention of a large spectrum of chronic diseases [11], and it may enact organ crosstalk through a positive immune modulation, which can ultimately result in a protective first-line of defense against viral infections, even including COVID-19 [12]. Physical activity has, in addition, immediate mood-enhancing and long-term positive psychological effects, which also makes it highly relevant during a condition of social confinement [13].

In light of the above, understanding the motives that guide people to enact physical activity has major public health implications. In this sense, a theoretical framework that evaluated human behavior, also considering physical activity, exercise, and sports, e.g., [14], is the self-determination theory (SDT) [15]. SDT starts from the implication that intentional human behavior can be assessed considering two main motivations: the autonomous motivation, which presumes that people engage in physical activity for personally endorsed reasons, and out of a sense of choice and volition; and the controlled motivation, which reflects acting for external motives, such as to gain a reward or to avoid punishment, guilt, or shame [16,17]. Some literature has demonstrated how autonomous motivation is associated with perseverance in physical activity contexts, e.g., [14]. In order to better comprehend the factors involved in future deliberation over behavior, social-cognitive models, e.g., [18] define behavioral intention as a motivational construct, reflecting the degree of deliberative planning and effort that an individual is prepared to invest in pursuing a given target behavior [19]. Hagger and colleagues [16] showed that behavioral intention mediates the relationship between higher levels of autonomous motivation and the implementation of physical activity. Accordingly, such results demonstrate that high levels of autonomous motivation permit people to shape their intentions in accordance with their motives.

Given the particular context that Italians experienced during the COVID-19 home confinement and the perception of threats related to the spread of this disease, a factor that appeared to have a relevant role in the relationship between motivation and physical activity behaviors was anxiety [20]. A longitudinal study conducted by Rogers et al. [21] analyzed the changes in physical activity during the lockdown period, showing that a general decrease in physical activity was associated with higher levels of anxiety. The inverse association of physical activity with anxiety is widely documented in the literature, both in regard to the pre-pandemic context [22] and during the lockdown period [21]. Other studies showed an inverse relationship between anxiety and healthy behaviors, including physical activity [23]. Although the relationship between anxiety and physical activity behavior is not clear, some scholars, considering other contexts, tried to evaluate the possible role of anxiety as a moderator between motivation and behavior, e.g., [24]. In sport or exercise contexts, the moderation role of anxiety between motives and behavior is quite elusive and less studied. In our previous study [20] we evaluated an integrated model linking both SDT variables and anxiety in relation to physical activity. To the best of our knowledge, there are no longitudinal studies regarding the pandemic period that evaluate the determinants of physical activity behavior, or consider the role of anxiety.

Given the dearth of studies investigating the moderating role of anxiety, the purpose of the present research was to explore the role of the anxiety as part of a mediation moderated model in a longitudinal design, assuming that the intention to do physical activity would positively mediate the relation between autonomous motivation and physical activity behavior. The hypothesized model is depicted in Figure 1.

## 2. Materials and Methods

### 2.1. Participants and Procedure

The study adopted a prospective design with two data collection occasions separated by about 3 weeks. Data were collected via online surveys written in the Italian language and administered between 17 March 2020 and 22 March 2020 (first occasion) and from 9 April 2020 to 24 April 2020 (second occasion). On both occasions, we spread the survey, following the snowball sampling procedure, and posted the link to the online survey on the main social media. We included only Italian participants over the age of 18. 

For the first data collection, participants completed questionnaires comprising self-report measures of autonomous motivation and intention, both towards physical activity behavior, and a measure of state anxiety. On the second occasion, participants self-reported their behavior regards physical activity during the quarantine period. All participants were recruited using online advertisements and informed of the general purpose of the research and their rights to anonymity and provided written informed consent before participating in the study. The time needed to complete the survey took approximately 10 min. Collected data were coded and processed anonymously. The Department of Psychology of Development and Socialization Processes Ethical Committee of the Sapienza University of Rome approved the study (protocol code: 727, 4 March 2019).

The number of participants who responded to our survey was 86. The demographic, descriptive characteristics and descriptive statistics of the sample are shown in Appendix A (available in the online Appendix A: https://osf.io/f7tha/ (accessed on 10 March 2022).

### 2.2. Measures

Autonomous Motivation. The relative degree of autonomous motivation was measured using a short form of the Behavioral Regulation in Exercise Questionnaire, version 3 (BREQ-3; 18 items) [25]. Participants were asked to answer using a 5-point Likert type scale (0 = “not true for me” and 4 = “very true for me”). In order to maximize the parsimony of the model in our study, the relative autonomy index (RAI) [26] was calculated. RAI is a single score derived from the subscales that give an index of the degree to which respondents feel self-determined. Higher, positive scores indicate greater relative autonomy; lower, negative scores indicate more controlled regulation. We used the Italian validation, which showed in the original version a six-factor structure and high reliability (ω = from 0.65 to 0.94) [27].

Intention. Intention was measured using four items (e.g., “I intend to do physical activity during the quarantine period”), answered using a 7-point Likert type scale (1 = “strongly disagree” and 7 = “strongly agree”). Item scores were aggregated into a single score, for which higher values indicated a greater intention toward the behavior. We used the Italian validation that showed, in a previous study, high reliability (α = 0.98) [20,28,29].

Anxiety was measured using the six-item short form of the State-Trait Anxiety Inventory (STAI) [30]. Participants were asked how they felt during the last week (e.g., “I feel worried”) and answered using a 6-point Likert type scale (1 = “never” and 6 = “always”). We used the Italian validated items [31]. The original version of this measure showed high reliability (α = 0.82).

*Self-reported behavior* was measured on both data collection occasions, considering the frequency in terms of weekly hours spent on physical activity during the quarantine period.

### 2.3. Data Analysis

We performed the data analysis using SPSS v. 27 (IBM Corp, 2020; Armonk, NY, USA) employing a statistical significance at α = 0.05. Descriptive analyses were calculated to describe the sample characteristics (i.e., sociodemographic) and the sample scores of autonomous motivation, intention, anxiety, and physical activity. We conducted a power analysis, the minimum sample size generated was *n* = 83, and the criteria that produced this estimate were (a) a minimum absolute significant path coefficient of 0.21, with a significance level used for hypothesis testing of *p* = 0.05, and (b) a power level of 0.60 (WARP PLS v. 7.0). Pearson correlations were performed to compute the bivariate correlations between the key variables of the study. The reliability of the measures was assessed using Cronbach’s alpha (α). Reliability was considered “excellent” for values of Cronbach’s α ≥ 0.90, “good” for α between 0.90 and 0.80, and “acceptable” for α between 0.80 and 0.70 [32]. Concerning the hypothesized model, we firstly tested the mediation model, assessing intention as the mediator in the relationship between autonomous motivation and physical activity, using the PROCESS macro [33] with 5000 bootstrap samples and unstandardized regression coefficients (model 4) [33]. Secondly, the moderated mediation model was assessed using model 59 of the PROCESS macro [33]. Our final hypothesized model (Figure 1) was a 1-mediator model, in which intention mediated the relationship between autonomous motivation and physical activity behavior. Furthermore, we tested the role of anxiety as a moderator of both the direct and indirect effects of autonomous motivation on physical activity, using a mean centering approach. The significance of the mediating and moderating effects was ascertained using bootstrap procedures with 5000 samples, following recent recommendations. All the effects were evaluated through 95% bias-corrected confidence intervals (CIs). Effects were considered significant if their confidence intervals did not include zero. In the case of significant interaction effects, we applied an extension of the Johnson–Neyman technique to assess the regions of significance. We also tested the effect of past physical activity behavior, measured in the first data collection occasion, on all the variables, as a statistical control, e.g., [34].

## 3. Results

### 3.1. Preliminary Results

All the measures showed a good reliability index, and the results show that autonomous motivation is positively and significantly correlated with both intention and physical activity behavior (r = 0.445, *p* < 0.001; r = 0.514, *p* < 0.001), intention is positively related to physical activity (r = 0.413, *p* < 0.001), while anxiety is not related to any of the model variables; see for descriptive statistics and correlation matrix (available in the online Appendix A: https://osf.io/f7tha/, see Appendix A, accessed on 10 March 2022).

### 3.2. Moderated Mediation Model

The moderated mediation model analysis was tested according to Hayes recommendations [33]. Specifically, the results of this model are summarized in Table 1.

In accordance with our hypotheses, autonomous motivation positively and significantly predicted both intention and physical activity (b = 0.58, *p* < 0.001; b = 0.17, *p* < 0.001). Subsequently, intention predicted physical activity (b = 0.08, *p* < 0.05). Concerning the influence of anxiety on the mentioned direct relations, it exerts an interactive effect on the autonomous motivation on physical activity behavior (b = −0.034, *p* < 0.001), thus, the conditional direct effect of anxiety on physical activity was significant in the low (1 SD below the mean) anxiety condition (b = 0.33, *p* < 0.001), while, on the contrary, in the high (1 SD above the mean) anxiety condition this effect became non-significant (*p* = 0.398; Figure 2). Furthermore, anxiety significantly moderated the relation between the intention on physical activity (b = 0.01, *p* < 0.05), and the effect was significant only in the high anxiety condition (b = 0.15, *p* < 0.001; Figure 3. For more details, please see Appendix A: https://osf.io/f7tha/, accessed on 10 March 2022).

Finally, the indirect (and conditional) effects of autonomous motivation on physical activity via the intention (available in the online Appendix A: https://osf.io/f7tha/, see Appendix A, accessed on 10 March 2022) were significant and positive only at the high levels of anxiety conditions (1 SD above the mean; b = 0.10, 95% CI: [0.014, 0.199]).

The inclusion of the past physical activity behavior (model 88 of the PROCESS macro) as a statistical control [34] led to a reduction in the effect of autonomous motivation on intention and on physical activity, while the relation between intention and physical activity became non-significant. In addition, the inclusion of past physical activity behavior led to an increase in the variance explained by the model for the physical activity during the quarantine period (for a full overview of the differences of all the effects of past behavior on all the variables, see the Appendix A: https://osf.io/f7tha/, see Appendix A, accessed on 10 March 2022). 

## 4. Discussion

The COVID-19 pandemic had a bi-directional hit on physical activity practice. On one hand, it has been shown that, during the lockdown period, the amount of time healthy adults spent in physical activity decreased by 60% while sedentary behaviors increased by 42% [35]. On the other hand, López-Sánchez et al. [35] observed that people decreased both incidental and planned physical activity under the circumstances of social distancing. The real root causes of the drop in physical activity levels during these exceptional circumstances have not yet been fully elucidated. It is likely that a multifactorial approach, including psychological pathways, represents the best option to understand this behavior. In this regard, the main purpose of the present study was to explore the role of anxiety as part of a mediation moderated model in a longitudinal design, considering that the intention to do physical activity positively mediates the relation between autonomous motivation and physical activity behavior.

The hypothesized model was supported, and the result showed that both autonomous motivation and intention are direct predictors of physical activity. Such results are in line with the literature describing the positive role of motivational and volitional aspects in determining the behavioral intention to practice physical activity and, moreover, to predict the implementation of this target behavior [16,28]. Indeed, scholars demonstrated that when people do physical activity for pleasure, interest, and personal satisfaction (i.e., autonomous motivation), they are more likely to fulfill their intentions to exercise and maintain this behavior, over time [16]. A similar trend has been found in our sample, within a COVID-19 pandemic context characterized by a period of lockdown and social distancing. Even if the literature showed that these two containment measures were fundamental to bending the curve of COVID-19 infections in Italy [36], these measures had a cost in terms of psychological health, such as high levels of anxiety, e.g., [37]. For this reason, we explored how anxiety affects the implementation of physical activity in a COVID-19 context, while also considering the motivational and social-cognitive aspects. Interestingly, anxiety acted as a moderator in the relation between autonomous motivation and physical activity behavior. In detail, the direct effect of autonomous motivation on the implementation of the behavior was stronger in participants with low anxiety than in people with high levels. Moreover, the results show that high levels of anxiety boost the intention effect of enacting this behavior. In other words, anxiety may boost physical activity depending on its levels, which moderate the reciprocal effects as outlined.

To speculate, our results show that anxiety influences the adoption of physical activity behavior through two different pathways: in the first, participants with low anxiety are keener to keep their physical routine, satisfying their needs as a direct effect of the pleasure to enact the behavior; while in the second, a high level of anxiety seems to “activate” a more volitional–intentional behavior, most likely as a coping strategy. In fact, the literature shows that individuals may adopt physical activity in order to reduce anxiety levels, bringing pleasure and relief [38]. 

To our knowledge, our study is the first longitudinal research that shows the psychosocial mechanisms involved in the practice of physical activity during the COVID-19 lockdown, considering the role of anxiety as a moderator of the relations between motivation, intention, and physical activity behavior. We also evaluated the role of anxiety in such a peculiar context, being predictive of physical activity practice. These strengths notwithstanding, the present research has a few inherent limitations. In the first place, the administration of the web survey sets out a caveat concerning the accessibility to an internet connection, and the possibility of participating in the survey [39]. Secondly, given that our findings are based on small sample size, the results from the present analyses should thus be treated with caution in terms of the generalizability of the findings.

## 5. Conclusions

In conclusion, given the absolute impact of lifestyle behaviors on mental health and quality of life, a multidisciplinary approach should be envisaged to understand the mechanisms explaining the rates of psychological issues (e.g., anxiety), and to develop the best methods and/or infrastructures to permit people to adhere to coping behaviors such as physical activity, or boosting autonomous behavior. This should be our immediate research priority to support both the general population and vulnerable groups, as a front line against this health emergency.

## Figures and Tables

**Figure 1 ijerph-19-04689-f001:**
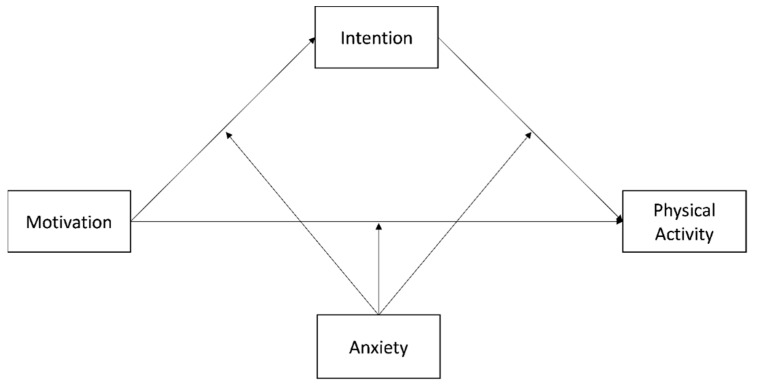
Graphic of the hypothesized moderated mediation model.

**Figure 2 ijerph-19-04689-f002:**
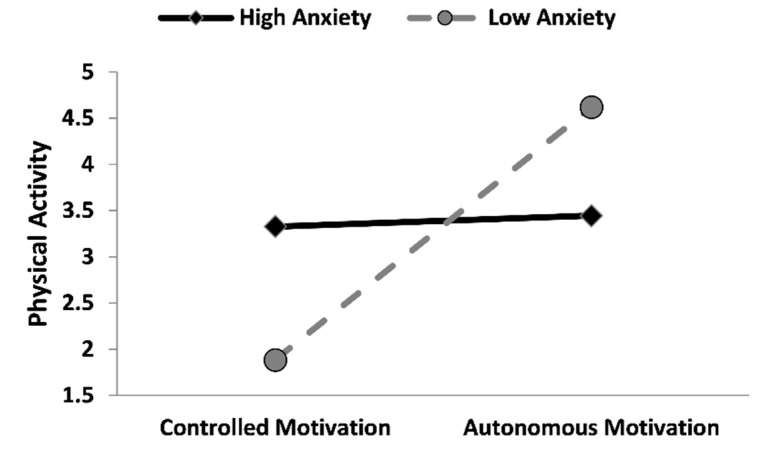
Simple slopes graph showing the moderating effect of anxiety on the relationship between autonomous motivation and days of physical activity during the week. Note. Lines were plotted using the mean sample scores of autonomous motivation and anxiety ± 1 SD.

**Figure 3 ijerph-19-04689-f003:**
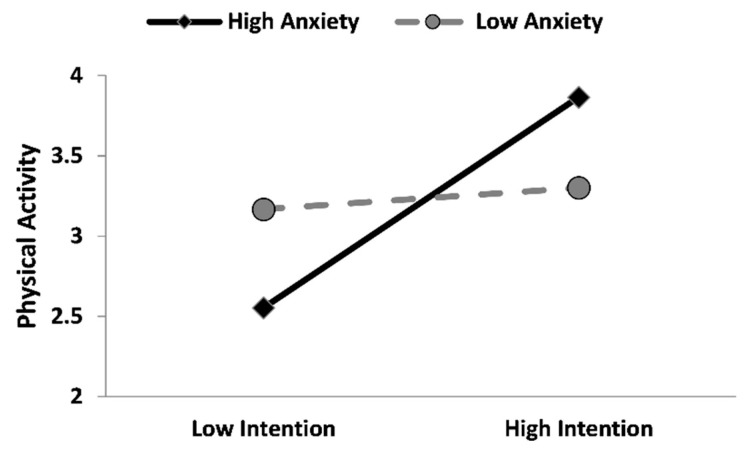
Simple slopes graph showing the moderating effect of anxiety on the relationship between intention and weekly days of physical activity. Note. Lines were plotted using the mean sample scores of autonomous motivation and anxiety ± 1 SD.

**Table 1 ijerph-19-04689-t001:** Coefficients for the moderated mediation model without past physical activity.

	Coeff.	SE	LLCI	ULCI
Intention				
Motivation	0.587	0.131	0.324	0.848
Anxiety	0.001	0.119	−0.235	0.237
Motivation *x* anxiety	0.021	0.028	−0.034	0.077
		R^2^ = 0.210		
Physical Activity				
Motivation	0.172	0.041	0.090	0.255
Intention	0.084	0.031	0.022	0.146
Anxiety	0.014	0.033	−0.052	0.081
Motivation *x* anxiety	−0.034	0.009	−0.052	−0.016
Intention *x* anxiety	0.015	0.006	0.001	0.028
		R^2^ = 0.415		

Note. N = 86; Coeff. = Coefficients; SE = standard error; LLCI = lower limit of the 95% confidence interval; ULCI = upper limit of the 95% confidence interval.

## Data Availability

Data are available under request to the corresponding author.

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
