# Peer review of "Anxiety, Motives, and Intention for Physical Activity during the Italian COVID-19 Lockdown: An Observational Longitudinal Study"

_ijerph, 2022, doi:10.3390/ijerph19084689_

Round 1

Reviewer 1 Report

Manuscript ID: ijerph-1655453

Title: Anxiety, motives, and intention for physical activity during the Italian COVID-19 lockdown: a longitudinal study

Thank you for providing a chance to review this manuscript.

Major revision

Title

    Page 1, line 3: As a longitudinal study, is there a control group? I didn't see it in the article.

Introduction

Page 1-2, line 41-77: This paragraph is too complicated and covers a lot of content. It can be summarized in some more concise and specific sentences.

Page 3, line 110-112: It has been nearly two years since the COVID-19 pandemic. What’s the innovation of this research? Is there any other similar research?

Page 3, line 113-117: Whether the theoretical support for the hypothetical model is sufficient.

Materials and Methods

Participants and Procedure

Page 3, line 126-132: There are no clear inclusion and exclusion criteria in the participant’ s section, how are the study subjects selected?

Page 3, line 133-135: I notice that you have passed the ethics review, but there is no ethics review batch number.

  Measures

Page 3, line 136: “Mesures” has a spelling error, it should be “Measures”.

Page 4, line 137-158: Whether Italy is the original country that used the scale, and if not, what is the reliability and validity of the original scale and the translated version, whether the original scale and the translated version are cited at the same time. Meanwhile, how reliable and valid is the scale in this study?

Data analysis

Page 4, line 160-182: Whether the data analysis was carried out by the STROBE guidelines1, 2? And data analysis section can be expressed in a more refined and organized sentence.

Results

 Participants

Page 4, line 185: Considering the BREQ-3 scale and STAI scale were used in the paper, is the sample size of n=86 sufficient? The choice of sample size should have a theoretical basis and the judging standard for valid samples should be stated.

Discussion

Page 7, line 260-265: Figures and tables belong to the “results” part and should not appear in the “discussion”. Reasonably arrange the presentation of the figures and tables in the results of the text.

Strengths and limitations

Page 7, line 273-283: This subtitle could be removed since your strengths and limitations contain limited content. Furthermore, strengths and limitations could be combined into one paragraph.

Taking all the comments above into consideration, this paper is decent and written with convincing data, but minor issues may still need to be reconsidered. And it is of great importance to request a native English speaker to check your writing and make the expression more understandable.

My best,

Your reviewer

References

  1. Cuschieri S. The STROBE guidelines. Saudi J Anaesth. Apr 2019;13(Suppl 1):S31-s34. doi:10.4103/sja.SJA_543_18
  2. Vandenbroucke JP, von Elm E, Altman DG, et al. Strengthening the Reporting of Observational Studies in Epidemiology (STROBE): explanation and elaboration. PLoS Med. Oct 16 2007;4(10):e297. doi:10.1371/journal.pmed.0040297

Reviewer 2 Report

Dear Authors,

This paper aimed evaluate the relation between autonomous motivation and physical activity, considering the role of behavioral intention and anxiety in a longitudinal moderated mediation model. There are some specific changes and suggestions that should be made to improve the quality of the paper.

General questions
- The objective in the summary must be inserted in the background and not in the methods.
Please correct.
- Pg01Ln29 – What does SD mean? The first time it appears in the document, it must be
written in full and only after that the abbreviation is used. To correct.
- Pg01Ln29 – What does F mean? The same as the previous one.
- Pg02Ln 91 – The reference is badly formatted. To correct.
- Insert in the participants and procedures section the number of participants who
participated in the study, as well as some anthropometric data (eg average weight, height,
age, etc.)
- What were the inclusion criteria of the participants for the study?
- Pg04Ln160 - Insert country and city from SPSS statistical program software?

- In the statistical analysis section, insert the level of significance used.
- Pg04Ln185 - This information “Participants who responded to our survey were 86”
should be in the methods section, more specifically in the participants and procedures
section.
- Put the “p” in italics.
- The first paragraph of the discussion section should remind you of the purpose of the
study. Please insert this information.

Best regards

Reviewer 3 Report

Review

Thank you for the possibility to review the article entitled: Anxiety, motives, and intention for physical activity during the Italian COVID-19 lockdown: a longitudinal study

I congratulate Authors on taking up an important and interesting topic.  

My comments to Authors.

Introduction

This section introduces the reader to the discussed topic very well.

I suggest unifying the aim of the work in the abstract and the main text.

Material and methods. 

This section is well described.  

Please fill in the information - where exactly was the survey questionnaire posted?

Results

According to rules titles of figures should be placed BELOW the figures.

In supplementary material - Table A: Sample characteristics would be worth adding numbers (not only % data)

Conclusions

Conclusions are supported by the results obtained by the Authors.

The work is interesting and carefully prepared and is suitable for publication after a small correction.

Round 2

Reviewer 1 Report

Good work, well done!

Reviewer 2 Report

Dear Authors,

I would like to congratulate the authors for the excellent work developed and for the improvement of the manuscript.

Best regards